# Self-Supervised Geometric Correspondence for Category-Level 6D Object Pose Estimation in the Wild

**Kaifeng Zhang**[1†]**, Yang Fu**[2]**, Shubhankar Borse**[3]**, Hong Cai**[3]**, Fatih Porikli**[3]**, Xiaolong Wang**[2]
[1]Tsinghua University, [2]UC San Diego, [3]Qualcomm AI Research

## Abstract

While 6D object pose estimation has wide applications across computer vision and robotics, it remains far from being solved due to the lack of annotations. The problem becomes even more challenging when moving to category-level 6D pose, which requires generalization to unseen instances. Current approaches are restricted by leveraging annotations from simulation or collected from humans. In this paper, we overcome this barrier by introducing a self-supervised learning approach trained directly on large-scale real-world object videos for category-level 6D pose estimation in the wild. Our framework reconstructs the canonical 3D shape of an object category and learns dense correspondences between input images and the canonical shape via surface embedding. For training, we propose novel geometrical cycle-consistency losses which construct cycles across 2D-3D spaces, across different instances and different time steps. The learned correspondence can be applied for 6D pose estimation and other downstream tasks such as keypoint transfer. Surprisingly, our method, without any human annotations or simulators, can achieve on-par or even better performance than previous supervised or semi-supervised methods on in-the-wild images. Code and videos are available at https://kywind.github.io/self-pose.

## 1 Introduction

Object 6D pose estimation is a long-standing problem for computer vision and robotics. In instance-level 6D pose estimation, a model is trained to estimate the 6D pose for one single instance given its 3D shape template (He et al., 2020; Xiang et al., 2017; Oberweger et al., 2018). For generalizing to unseen objects and removing the requirement of 3D CAD templates, approaches for category-level 6D pose estimation are proposed (Wang et al., 2019b). However, learning a generalizable model requires a large amount of data and supervision. A common solution in most approaches (Wang et al., 2019b; Tian et al., 2020; Chen et al., 2020a; 2021; Lin et al., 2021) is leveraging both real-world (Wang et al., 2019b) and simulation labels (Wang et al., 2019b; Chang et al., 2015) at the same time for training. While there are limited labels from the real world given the high cost of 3D annotations, we can generate as many annotations as we want in simulation for free. However, it is very hard to model the large diversity of in-the-wild objects with a simulator, which introduces a large sim-to-real gap when transferring the model trained with synthetic data.

Although the real-world labels are hard to obtain, the large-scale object data is much more achievable (Fu & Wang, 2022). In this paper, we propose a self-supervised learning approach that directly trains on large-scale unlabeled object-centric videos for category-level 6D pose estimation. Our method does not require any 6D pose annotations from simulation or human labor for learning. This allows the trained model to generalize to in-the-wild data. Given a 3D object shape prior for each category, our model learns the 2D-3D dense correspondences between the input image pixels and the 3D points on the categorical shape prior, namely **geometric correspondence**. The object 6D pose can be solved with the correspondence pairs and the depth map using a pose fitting algorithm.

We propose a novel **Categorical Surface Embedding (CSE)** representation, which is a feature field defined over the surface of the categorical canonical object mesh. Every vertex of the canonical mesh is encoded into a feature embedding to form the CSE. Given an input image, we use an image encoder to extract the pixel features to the same embedding space. By computing the similarity

---

[†]Work done while an intern at UC San Diego.

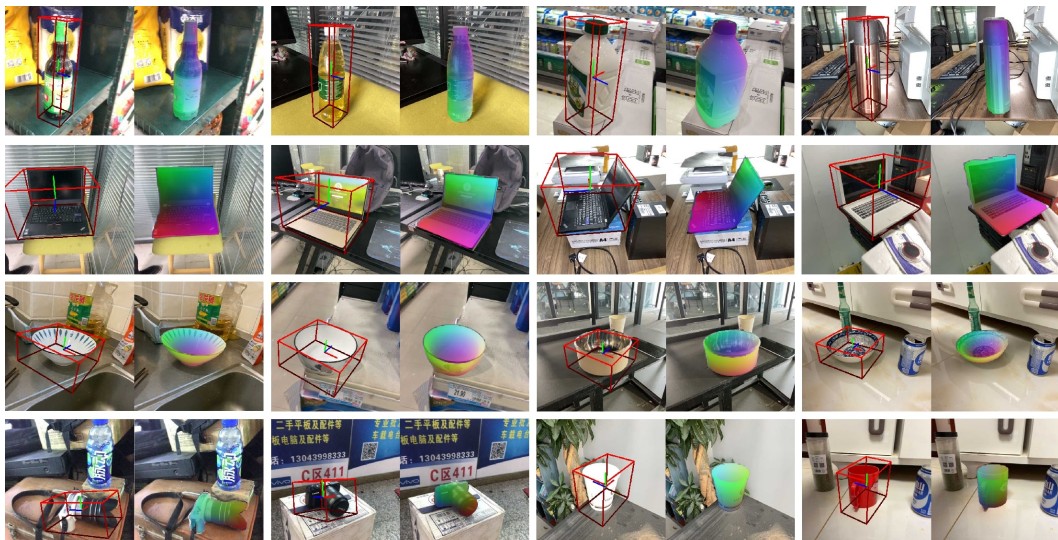

Figure 1: **Examples of self-supervised category-level 6D pose estimation in the wild.** We propose a novel Categorical Surface Embedding representation to learn categorical 2D-3D geometric correspondences. For each example, we visualize the object 6D pose and its correspondence map.

between the 3D vertex embeddings and the pixel embeddings, we can obtain the 2D-3D geometric correspondence. With such correspondence, we lift the 2D image texture to a 3D mesh, and project the textured mesh back to the 2D RGB object image, segmentation mask and depth using differentiable rendering. We use reconstruction losses by comparing them to the 2D ground-truths for training. However, the reconstruction tasks do not provide enough constraints on learning the high dimensional correspondence.

To facilitate the optimization in training, we propose novel losses that establish cycle-consistency across 2D and 3D space. Within a single instance, our network can estimate the 2D-3D dense correspondence $\phi$ using CSE and the global rigid transformation $\pi$ for projecting 3D shapes to 2D using another encoder. Given a 2D pixel, we can first find its corresponding 3D vertex with $\phi$ and then project it back to 2D with $\pi$, the projected location should be consistent with the starting pixel location. We can design a similar loss by forming the cycle starting from the 3D vertex. This provides an **instance cycle-consistency loss** for training. Beyond a single instance, we also design cycles that go across different object instances within the same category. Assuming given instance $A$ and $B$, this cycle will include a forward pass across the 3D space, and a backward pass across the 2D space: (i) Forward pass: Starting from one 2D pixel in instance $A$, we can find its corresponding 3D vertex using $\phi$. This 3D vertex can easily find its corresponding 3D point location in instance $B$ given the mesh is defined in canonical space. The located 3D point is then projected back to 2D using $\pi$ for instance $B$. (ii) Backward pass: We leverage the self-supervised pre-trained DINO feature (Caron et al., 2021) to find the 2D correspondence between two instances $A$ and $B$. The located 2D pixel in instance $B$ during the forward pass can find its corresponding location in instance $A$ using the 2D correspondence, which provides a **cross-instance cycle-consistency loss**. The same formulation can be easily extended to videos, where we take $A$ and $B$ as the same instance across time, and this additionally provides a **cross-time cycle-consistency loss**.

We conduct our experiments on the in-the-wild dataset Wild6D (Fu & Wang, 2022). Surprisingly, our self-supervised approach that directly trains on the unlabeled data performs on par or even better than state-of-the-art approaches which leverage both 3D annotations as well as simulation. We visualize the 6D pose estimation and the geometric correspondence map in Figure 1. Besides Wild6D, we also train and evaluate our model in the REAL275 (Wang et al., 2019b) dataset and shows competitive results with the fully supervised approaches. Finally, we evaluate the CSE representation on keypoint transfer tasks and achieve state-of-the-art results. We highlight our main contributions as follows:

- To the best of our knowledge, this is the first work that allows self-supervised 6D pose estimation training in the wild.
- We propose a framework that learns a novel Categorical Surface Embedding representation for 6D pose estimation.
- We propose novel cycle-consistency losses for training the Categorical Surface Embedding.

## 2 RELATED WORK

**Category-level 6D Pose Estimation.** Compared to instance-level 6D pose estimation (Xiang et al., 2017; Li et al., 2018; Wang et al., 2019a), the problem of category-level pose estimation is more under-constrained and challenging due to large intra-class shape variations. Wang et al. (2019b) propose the Normalized Object Coordinate Space (NOCS) which aligns different objects of the same category into a unified normalized 3D space. With the NOCS map estimation, the 6D pose is solved using the Umeyama algorithm (Umeyama, 1991). Following this line of research, subsequent works (Tian et al., 2020; Chen et al., 2020a; Fu & Wang, 2022) are proposed to learn more accurate NOCS representation. Aside from using NOCS, 6D pose estimation can also be approached by directly regressing the pose (Chen et al., 2021; Lin et al., 2021), estimating keypoint locations (Lin et al., 2022), and reconstructing the object followed by an alignment process (Irshad et al., 2022a;b). However, all these approaches require training on human-annotated real datasets (Wang et al., 2019b; Ahmadyan et al., 2021) or synthetic datasets (Wang et al., 2019b; Chang et al., 2015). Our work follows the same estimate-then-solve pipeline as NOCS, while doing this in a self-supervised manner without using synthetic data or human annotations on real data.

**Self-supervision for Pose Estimation.** With the high cost of getting 3D annotations, different self-supervised training signals are proposed for better generalization. One effective strategy is to perform the sim-to-real adaptation on the model pre-trained on the synthetic data (Chen et al., 2020b; He et al., 2022; Gao et al., 2020; You et al., 2022). Another line of research focuses on achieving better generalization ability by semi-supervised training on both synthetic datasets and unlabeled real data (Fu & Wang, 2022; Manhardt et al., 2020; Peng et al., 2022). All of these works require first training with the synthetic data, which can provide a good initialization for 6D pose estimation. To the best of our knowledge, our work is the first one that directly trains on in-the-wild images without any annotations. This avoids dealing with the sim2real gap problem in training and it has the potential to generalize to a larger scale of data without carefully designing our simulator.

**Self-supervised Mesh Reconstruction and Geometric Correspondence.** Our work is related to recent methods for mesh reconstruction by applying differentiable renderers (Kato et al., 2018; Liu et al., 2019) together with the predicted object shape, texture and pose (Kanazawa et al., 2018; Goel et al., 2020; Tulsiani et al., 2020; Li et al., 2022; Wu et al., 2021; Hu et al., 2021; Yang et al., 2021a; Ye et al., 2021). However, the object pose estimation results from these works are still far from satisfactory, especially for daily objects with complex structures and diverse appearances. Compared to these works, our method is more close to recent works on learning the 2D-3D geometric correspondence from RGB images in a self-supervised manner (Güler et al., 2018; Neverova et al., 2020; Kulkarni et al., 2019; 2020; Yang et al., 2021b). For instance, Yang et al. (2021b) establishes the dense correspondences between image pixels and 3D object geometry by predicting surface embedding for each vertex in the object mesh. However, the learned surface embeddings are object-specific and hard to generalize to different objects. To supervise correspondence learning, most existing works design the geometric consistency loss (e.g., Kulkarni et al. (2019)) within a single object. In this paper, we propose novel cross-instance and cross-time cycle-consistency losses that build a 4-step cycle and achieve better geometric correspondence.

**Learning from Cycle-consistency.** To construct the 4-step cycle, we will need to establish correspondence across images, which is related to visual correspondence learning in semantics (Rocco et al., 2017; Truong et al., 2021; Jeon et al., 2018; Caron et al., 2021; Huang et al., 2019) and for tracking (Vondrick et al., 2018; Wang et al., 2019c; Li et al., 2019; Xu & Wang, 2021; Jabri et al., 2020). For example, Wang et al. (2019c) proposes to perform cycle-consistency learning by tracking objects forward and backward in time. Different from these works, our method constructs cycles across 2D and 3D space. In this aspect, our work is more close to (Zhou et al., 2016), where 3D CAD models are used as intermediate representations to help improve 2D image matching. However, they are not learning the 3D model but using existing CAD model templates for learning better 2D keypoint transfer. Our work learns the 2D-3D geometric correspondence and the 3D structure jointly.

## 3 METHOD

To estimate the 6D pose, we propose to learn the Categorical Surface Embedding (CSE) for finding geometric correspondence (2D-3D correspondence) in a self-supervised manner. We perform learning under a reconstruction framework with differentiable rendering as shown in Fig. 2. Our model takes a single RGB image as input, with ground truth object mask and depth as supervision. Given the

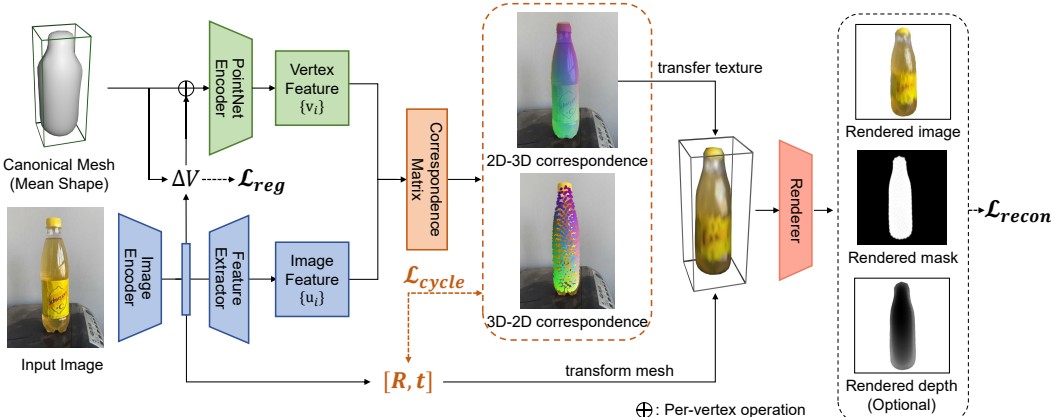

Figure 2: **Framework Overview**. Given the input image, the image encoder and feature extractor (blue) predict the canonical mesh deformations $\Delta V$, transformations $[\mathbf{R}|\mathbf{t}]$ and per-pixel features. The mesh encoder (green) predicts mesh per-vertex features. We compute the feature similarity matrix and bidirectional correspondences (orange), which are used to transfer image pixel colors to mesh texture. Finally, we use differentiable rendering to render the RGB image, segmentation mask and (optionally) depth map. The framework is trained with reconstruction loss $\mathcal{L}_{recon}$, cycle-consistency loss $\mathcal{L}_{cycle}$ and regularization loss $\mathcal{L}_{reg}$.

input RGB image, we compute the image feature in the space of CSE. At the same time, we adopt the PointNet (Qi et al., 2017) to encode the deformed categorical mesh shape (Section 3.1) and extract the vertex features. We compute the geometric correspondence between these two CSE feature maps and use the correspondence to transfer the texture from a 2D image to a 3D shape. We project the textured 3D shape with a regressed pose to the 2D image, mask, and depth via differentiable rendering. We compute the reconstruction losses against the 2D ground-truths (Section 3.2). However, reconstruction alone does not provide enough constraint for learning the high-dimensional geometric correspondence. We propose multi-level cycle-consistency losses (Section 3.3) to enforce the consistency of geometric correspondence across instances and time. We summarize our training and inference in Section 3.4.

### 3.1 CANONICAL SHAPE PRIOR DEFORMATION

The first step of our model is to learn a categorical canonical shape prior. We choose to use a triangular mesh $\mathbf{S} = \{\bar{\mathbf{V}}, \mathbf{F}\}$, where $\bar{\mathbf{V}} \in \mathbb{R}^{3 \times N}$ are canonical vertices and $\mathbf{F} \in \mathbb{R}^{3 \times M}$ are canonical faces that encode the mesh topology. We use $\Delta \mathbf{V}$ to model the deformation of each instance in the same category, and the shape of an instance can be represented as $\mathbf{V} = \bar{\mathbf{V}} + \Delta \mathbf{V}$, where $\bar{\mathbf{V}}$ is a categorical shape prior and $\Delta \mathbf{V}$ is the instance-specific deformation. The shape prior not only describes the approximate shape of the category, but also defines the canonical pose, such that the 6D pose estimation problem can turn into predicting the relative pose between the observed object and the canonical meshes. We initialize $\bar{\mathbf{V}}$ with a selected shape, and it remains learnable during training. Given an input image, we predict an implicit shape code $\mathbf{u}_{shape} \in \mathbb{R}^s$ with our image encoder. Then, for each vertex on the canonical mean mesh, we concatenate the vertex positions $[x_i, y_i, z_i]$ with the shape code $\mathbf{u}_{shape}$, and use a MLP $f_{shape} : \mathbb{R}^{s+3} \rightarrow \mathbb{R}^3$ to predict the per-vertex offset amount along 3 dimensions $x, y, z$ to obtain $\Delta \mathbf{V}$.

### 3.2 CATEGORICAL SURFACE EMBEDDING AND CORRESPONDENCE

Once we have the deformed mesh based on the input image, we can extract the vertex features using a PointNet (Qi et al., 2017) encoder (green box in Fig. 2) as $\{\mathbf{v}_1, \cdots, \mathbf{v}_N\} \in \mathbb{R}^{N \times d}$, where $N$ is the number of vertices and $d$ is embedding dimension. We name this per-vertex embedding as the Categorical Surface Embedding, for it introduces a shared embedding space across all mesh instances in the same category. On the image side, we use a feature extractor network to obtain the pixel-wise features $\{\mathbf{u}_1, \cdots, \mathbf{u}_{h \times w}\} \in \mathbb{R}^{h \times w \times d}$, where $h, w$ are the height and width.

After extracting image features and vertex features, we measure the cosine distance between per-pixel features and per-vertex features, from where we obtain the image-mesh and mesh-image correspondence matrices via a Softmax normalization over all mesh vertices and over all pixels:

$$W_{ij}^{\text{2D-3D}} = \frac{\exp\left(\cos\langle \mathbf{u}_i, \mathbf{v}_j\rangle / \tau\right)}{\sum_i \exp\left(\cos\langle \mathbf{u}_i, \mathbf{v}_j\rangle / \tau\right)}, \quad W_{ji}^{\text{3D-2D}} = \frac{\exp\left(\cos\langle \mathbf{u}_i, \mathbf{v}_j\rangle / \tau\right)}{\sum_j \exp\left(\cos\langle \mathbf{u}_i, \mathbf{v}_j\rangle / \tau\right)}. \quad (1)$$

$W_{ij}^{\text{2D-3D}}$ and $W_{ji}^{\text{3D-2D}}$ are entries in the correspondence matrices that represent the 2D-3D and 3D-2D correspondence values between mesh vertex $i$ and pixel $j$. $\tau$ is the temperature parameter. With the matrices, we construct the 2D-3D correspondence mapping and 3D-2D correspondence mapping:

$$\mathbf{q}_j = \sum_{i=1}^{N} W_{ij}^{\text{2D-3D}} \cdot [x_i, y_i, z_i], \quad \mathbf{p}_i = \sum_{j=1}^{h \times w} W_{ji}^{\text{3D-2D}} \cdot [X_j, Y_j]. \tag{2}$$

$\mathbf{q}_j$ and $\mathbf{p}_i$ represent the corresponding 3D location of pixel $j$ and the corresponding 2D location of vertex $i$, respectively. We denote the 2D-3D mapping as $\phi : \mathbb{R}^2 \to \mathbb{R}^3$ such that $\phi([X_j, Y_j]) = \mathbf{q}_j$, and the 3D-2D mapping as $\psi : \mathbb{R}^3 \to \mathbb{R}^2$ such that $\psi([x_i, y_i, z_i]) = \mathbf{p}_i$. We emphasize that our computation of correspondence is based on the similarities between per-vertex surface embedding and per-pixel image feature, which is different from previous work (Kulkarni et al., 2019) on direct repression of the 2D-3D mapping function as neural network outputs.

**Texture transfer.** The geometric correspondence can serve as texture flow that transfers image pixel colors to the mesh. Specifically, we propose to perform texture transfer using the predicted 3D-2D correspondence, which leads to a textured mesh: $\text{tex}_i = \sum_{j=1}^{h \times w} W_{ji}^{\text{3D-2D}} \cdot I_j$, where $\text{tex}_i$ is the texture color at mesh vertex $i$, and $I_j$ is the RGB value at image pixel $j$.

**Reconstruction Loss.** We perform reconstruction in 2D space for supervision. We first estimate a mesh transformation $[\mathbf{R}|\mathbf{t}]$ ($\mathbf{R} \in SO(3), \mathbf{t} \in \mathbb{R}^3$) using the image encoder. We use the continuous 6D representation for 3D rotation (Zhou et al., 2019), and 3D translation. After transforming the mesh with the estimated rotation and translation, we adopt SoftRas (Liu et al., 2019) (deep orange box in Fig. 2) as the differentiable renderer to generate the RGB image, segmentation mask and depth map. The reconstruction loss is defined as follows:

$$\mathcal{L}_{\text{recon}} = \beta_{\text{texture}} \cdot \|\hat{I} \cdot \hat{S} - I \cdot S\|_2^2 + \beta_{\text{mask}} \cdot \|\hat{S} - S\|_2 + \beta_{\text{depth}} \cdot \|\hat{D} \cdot \hat{S} - D \cdot S\|_2^2 \tag{3}$$

where $I, S, D$ are the input image, ground truth segmentation mask, and the depth map, respectively, and $\hat{I}, \hat{S}, \hat{D}$ are the corresponding rendering results. With the reconstruction loss, our model can learn shape deformation and mesh transformation with back-propagation. The texture loss further enforces the color consistency between the projected vertices and the pixels. However, color consistency is not sufficient to learn fine-grained, high-dimensional correspondence. To this end, in the next section, we introduce the novel cycle-consistency loss which learns a coordinate-level correspondence based on the reconstructed shape and pose.

### 3.3 LEARNING WITH CYCLE-CONSISTENCY

We propose multiple cycle-consistency losses across 2D and 3D space for learning, including constructing a cycle within a single instance, and cycles across different instances and time.

**Instance cycle-consistency.** In Eq.2 we introduced the correspondence formulation, the corresponding 2D-3D mapping $\phi : \mathbb{R}^2 \to \mathbb{R}^3$ and 3D-2D mapping $\psi : \mathbb{R}^3 \to \mathbb{R}^2$. We also introduced the mesh transformation $[\mathbf{R}|\mathbf{t}]$ regressed by the image encoder. For instance cycle-consistency, we encourage the consistency between the predicted mapping and the camera projection. In an ideal situation, $\phi$ should be consistent with the camera projection (which we denote as $\pi = \mathbf{C}[\mathbf{R}|\mathbf{t}]$, $\mathbf{C}$ is the camera matrix), and $\psi$ should be consistent with the inverse projection $\pi^{-1}$. Naturally, this leads to 2 cycle losses, one combining $\phi$ with $\pi$, the other combining $\psi$ with $\pi^{-1}$ (See Fig.3 left for an illustration):

$$\mathcal{L}_{\text{2D-3D}} = \frac{1}{|I|} \sum_{\mathbf{p} \in I} \|\pi(\phi(\mathbf{p})) - \mathbf{p}\|_2^2, \quad \mathcal{L}_{\text{3D-2D}} = \frac{1}{|V|} \sum_{\mathbf{q} \in \mathbf{V}} \|\pi^{-1}(\psi(\mathbf{q})) - \mathbf{q}\|_2^2. \tag{4}$$

We incorporate a visibility constraint into this loss by only computing cycles that involve non-occluded vertices under the estimated projection. (See Appendix A.2 for more details.)

**Cross-instance and cross-time cycle-consistency.** To handle the large variance of object shape and appearance across instances in the same category, we design a loss that encourages the consistency of the geometric correspondence across instances. We propose a cross-instance cycle-consistency loss as illustrated in Fig.3. The loss is built by a 4-step cycle, consisting of 2 images (i.e., source and target images) and their respective reconstructed meshes. It contains a forward pass through the 3D space, and a backward pass through the 2D space. In the forward pass, given a pixel $\mathbf{p}_i$ in the source image, we first map the point to the 3D mesh via the predicted correspondence $W^{\text{2D-3D}}$, creating a correspondence distribution over the source mesh surface. In this distribution, the value at

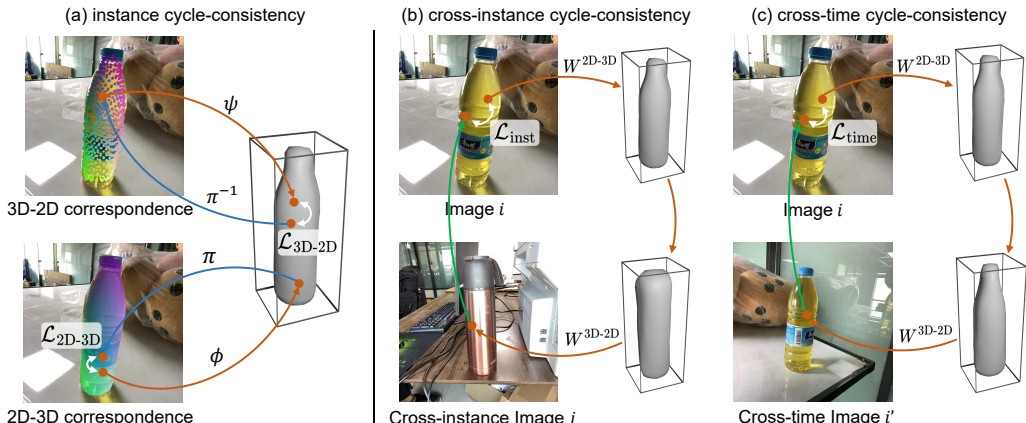

Figure 3: **An illustration of the cycle-consistency losses.** Left: instance cycle-consistency. Loss is defined by re-projection offsets between the image and the reconstructed mesh, using 3D-2D and 2D-3D mappings (brown) and estimated projection (blue). Middle and right: the cross-instance cycle-consistency and cross-time cycle-consistency, established as a 4-step cycle, using our learned correspondence (brown) and directly-extracted DINO correspondence pairs (green).

each vertex indicates the correspondence intensity with $\mathbf{p}_i$. Then, we directly transfer the distribution on the source mesh onto the target mesh using the one-to-one vertex correspondence induced by deformation (recall that both meshes are deformed from the same mesh prior). We then re-project the correspondence distribution back from the target mesh to the target image using $W^{\text{3D-2D}}$, resulting in a distribution over target image pixels. Concretely, this distribution equals the product of two correspondence matrices, from where we can calculate the correspondence:

$$W^{\text{forward}} = W^{\text{2D-3D}}_{\text{src}} \cdot W^{\text{3D-2D}}_{\text{tgt}}, \qquad \mathbf{q}^{\text{forward}}_i = \sum_{i'=1}^{h \times w} \frac{W^{\text{forward}}_{ii'}}{\sum_j W^{\text{forward}}_{ij}} \cdot [X_{i'}, Y_{i'}], \tag{5}$$

where we calculate $\mathbf{q}^{\text{forward}}_i$ as the corresponding pixel on the target image for $\mathbf{p}_i$ by a weighted sum over the locations of target pixels. This constructs the forward correspondence mapping $f_{\text{forward}} : \mathbf{p}_i \to \mathbf{q}^{\text{forward}}_i$. To form a cycle, we need to connect it with a backward mapping $f_{\text{backward}}$ which operates directly between the source image and the target image and finds correspondences. To achieve this, we leverage a self-supervised pretrained feature extraction network DINO (Caron et al., 2021), and follow a prior work (Goodwin et al., 2022) to extract $k$ high-confidential correspondence pairs between the target and source image, $\{(\mathbf{q}^{\text{forward}}_i, \mathbf{p}^{\text{cycle}}_i)\}_{i=1}^k$, where $\mathbf{p}^{\text{cycle}}_i$ is the pixel location going through the whole cycle back in the source image. Our cross-instance cycle-consistency loss $\mathcal{L}_{\text{inst}}$ is defined to be the L2 distance on position offsets over the $k$ DINO correspondence pairs:

$$\mathcal{L}_{\text{inst}} = \frac{1}{k} \sum_{i=1}^k \|\mathbf{p}^{\text{cycle}}_i - \mathbf{p}_i\|_2^2. \tag{6}$$

$\mathcal{L}_{\text{inst}}$ operates on images of different object instances. This can be easily extended to a cross-time setting by selecting source and target images to be different frames of the same video clip, leveraging the continuity in time to learn geometric consistency. We denote this **cross-time consistency loss** as $\mathcal{L}_{\text{time}}$. An illustration of both losses can be found in Fig.3. We combine all cycle-consistency losses as $\mathcal{L}_{\text{cycle}} = \beta_{\text{2D-3D}} \cdot \mathcal{L}_{\text{2D-3D}} + \beta_{\text{3D-2D}} \cdot \mathcal{L}_{\text{3D-2D}} + \beta_{\text{inst}} \cdot \mathcal{L}_{\text{inst}} + \beta_{\text{time}} \cdot \mathcal{L}_{\text{time}}$.

## 3.4 TRAINING AND INFERENCE

**Training.** All the operations in our model are differentiable, which allows us to train all components in an end-to-end manner. Besides the reconstruction loss $\mathcal{L}_{\text{recon}}$ and the cycle-consistency loss $\mathcal{L}_{\text{cycle}}$, we also apply regularization loss $\mathcal{L}_{\text{reg}}$ on the learned shape to enforce shape smoothness, symmetry and minimize deformation. The total training objective is a weighted sum of them, $\mathcal{L}_{\text{total}} = \lambda_{\text{recon}} \cdot \mathcal{L}_{\text{recon}} + \lambda_{\text{cycle}} \cdot \mathcal{L}_{\text{cycle}} + \lambda_{\text{reg}} \cdot \mathcal{L}_{\text{reg}}$. (See Appendix A.2 for more details.)

**Inference.** During inference, our model can estimate object shape and correspondences with RGB images. Additionally, for inference of the 6D pose, depth is also required. We first extract the 2D-3D geometric correspondence mapping, forming a correspondence pair set of $M$ elements, where $M$ is

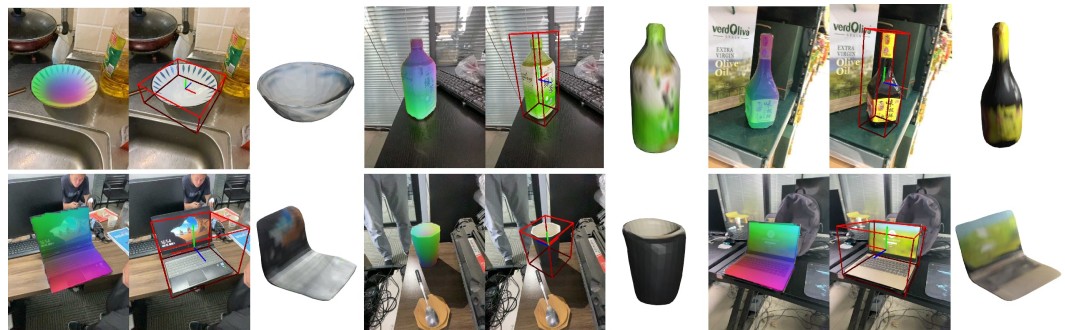

Figure 4: **Visualizing pose estimation and mesh reconstruction on Wild6D.** We show the correspondence mapping, the estimated pose and the reconstructed meshes.

| Methods | Data | $IOU_{0.25}$ | $IOU_{0.5}$ | 5 degree 2cm | 5 degree 5cm | 10 degree 2cm | 10 degree 5cm |
|---|---|---|---|---|---|---|---|
| CASS (Chen et al., 2020a) | C+R | 19.8 | 1.0 | 0 | 0 | 0 | 0 |
| Shape-Prior (Tian et al., 2020) | C+R | 55.5 | 32.5 | 2.6 | 3.5 | 9.7 | 13.9 |
| DualPoseNet (Lin et al., 2021) | C+R | 90.0 | 70.0 | 17.8 | 22.8 | 26.3 | 36.5 |
| GPV-Pose (Di et al., 2022) | C+R | 91.3 | 67.8 | 14.1 | 21.5 | 23.8 | 41.1 |
| RePoNet (Fu & Wang, 2022) | C+W* | 84.7 | **70.3** | 29.5 | 34.4 | 35.0 | 42.5 |
| Ours | W* | **92.3** | 68.2 | **32.7** | **35.3** | **38.3** | **45.3** |

Table 1: **Comparison with the SOTA methods on Wild6D**. The "Data" column records the data for training ,with C=CAMERA25, R=REAL275, W=Wild6D, "*"=not using pose annotation.

the number of pixels on the object foreground mask. To remove probable outliers, we first apply a selection scheme to the correspondence pairs based on the consistency between 2D-3D and 3D-2D correspondence. Concretely, we define a confidence measure based on the cycle-consistency of our 2D-3D and 3D-2D matching:

$$\text{conf}_p = \exp\left(\|\psi(\phi(p)) - p\|_2^2\right), \quad \forall p \in I. \tag{7}$$

High confidence indicates agreement between both directions of correspondence. We sort correspondence pairs using this confidence, and select those above a threshold $\alpha = 0.5$. We also lift the depth map of the object into a partial point cloud, forming a 3D rigid transformation estimation problem. We use the Umeyama algorithm (Umeyama, 1991) to solve this problem, with RANSAC (Fischler & Bolles, 1981) for outlier removal.

## 4 EXPERIMENTS

### 4.1 EXPERIMENT SETTINGS

**Category-level 6D object pose estimation.** We conduct the main body of experiments on Wild6D (Fu & Wang, 2022), which contains a rich collection of 5,166 videos across 1722 different objects and 5 categories (bottle, bowl, camera, laptop and mug) with diverse appearances and backgrounds. The training set of Wild6D provides RGB-D information and object foreground segmentation masks generated by Mask R-CNN (He et al., 2017), and the test set includes 6D pose labels generated by human annotation. On this dataset, we compare with the current state-of-the-art semi-supervised method, RePoNet (Fu & Wang, 2022) and several pretrained supervised methods transferred to this dataset. Following Wang et al. (2019b), we report the mean Average Precision (mAP) for different thresholds of the 3D Intersection over Union (IoU) and the $m$ degree, $n$ cm metric.

We also train and evaluate our model on the widely-used REAL275 dataset (Wang et al., 2019b), which contains a smaller set of videos (7 training scenes and 6 testing scenes), posing a large challenge on training the model. We categorize prior works on REAL275 into 3 categories based on supervision: 1) supervised; 2) self-supervised (with synthetic data and annotations); 3) self-supervised (without synthetic data). Our method tackles the problem in the most challenging setting.

**Keypoint transfer.** To thoroughly evaluate our novel CSE representation and 2D-3D geometric correspondence, we evaluate on the keypoint transfer (KT) task. We use the CUB-200-2011 dataset (Wah et al., 2011) containing 6,000 training and test images of 200 species of birds. Each bird is annotated with foreground segmentation and 14 keypoints. During training, we use only the RGB

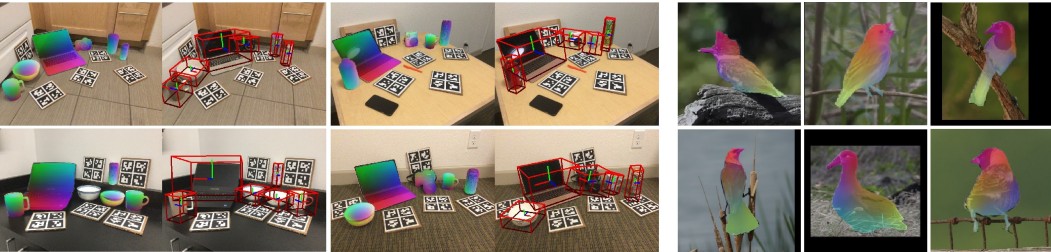

Figure 5: **Visualizing pose estimation and correspondence on REAL275 (left) and CUB-200-2011 (right).**

| Supervision | Methods | Data | $IOU_{0.25}$ | $IOU_{0.5}$ | 5 degree 5cm | 10 degree 5cm |
|---|---|---|---|---|---|---|
| Supervised | NOCS (Wang et al., 2019b) | C+R | 84.8 | 78.0 | 10.0 | 25.2 |
| | Shape-Prior (Tian et al., 2020) | C+R | - | 77.3 | 21.4 | 54.1 |
| | FS-Net (Chen et al., 2021) | C+R | **95.1** | **92.2** | 28.2 | 60.8 |
| | DualPoseNet (Lin et al., 2021) | C+R | - | 79.8 | **35.9** | **68.8** |
| Self-supervised (with synthetic data and annotations) | Peng et al. (2022) | C+R* | 83.2 | 73.0 | 19.6 | 54.5 |
| | RePoNet (Fu & Wang, 2022) | C+R* | **85.8** | 76.9 | 31.3 | 56.8 |
| | UDA-COPE (Lee et al., 2022) | C+R* | 84.0 | **82.6** | **34.8** | **66.0** |
| | Chen et al. (2020b) | S | 15.5 | 1.3 | 0.9 | 2.4 |
| | CPPF (You et al., 2022) | S | 78.2 | 26.4 | 16.9 | 44.9 |
| Self-supervised (without synthetic data) | He et al. (2022) | R*+Y* | 83.5 | **58.7** | 5.6 | 17.4 |
| | Ours-REAL | R* | 76.3 | 41.7 | 11.6 | 28.3 |
| | Ours-Wild6D | W* | **89.3** | 49.5 | **13.7** | **33.7** |

Table 2: **Comparison with SOTA methods on REAL275**. In "Data" column, C=CAMERA25, R=REAL275, S=synthetic objects, Y=YCB (Xiang et al., 2017), W=Wild6D, "*"=not using pose annotation.

images and segmentation maps, with no depth or keypoint annotation. For inference, following CSM (Kulkarni et al., 2019), we take the Percentage of Correct Keypoints (PCK) as the evaluation metric, which measures the percentage of correctly transferred keypoints from a source image to a target image if the keypoint is visible in both. Please refer to Appendix A.3 for more details.

## 4.2 COMPARISON WITH STATE-OF-THE-ART

**Pose estimation performance.** The pose estimation performance on Wild6D is reported in Table 1. Surprisingly, our method outperforms all previous methods only except for the $IoU_{0.5}$ metric, while being the only method that uses completely no synthetic data or 3D annotated data. On pose estimation metric 5 degree, 2cm and 10 degree, 2cm, our method outperforms the previous state-of-the-art semi-supervised method RePoNet (Fu & Wang, 2022) by 3.2% and 3.3% respectively. A visualization of the predicted correspondence, pose and the reconstructed mesh is shown in Fig.4. More visualizations are given in Fig. 1 and Appendix C.

The result on REAL275 is reported in Table 2. In the table, *Ours-REAL* is trained only on REAL275, and *Ours-Wild6D* is trained only on Wild6D. We outperform previous self-supervised methods on most metrics. Compared with supervised methods and self-supervised methods using annotated synthetic data, our performance is lower mainly because 1) the REAL275 training set is small in size, and 2) synthetic data with ground truth shape and pose can provide a strong supervision. Compared with *Ours-REAL275*, *Ours-Wild6D* achieves a higher performance, showing a strong generalization ability across image domains. A visualization of the correspondence and pose estimation results on REAL275 is shown in Fig.5.

**Keypoint Transfer Performance.** Our keypoint transfer results on the CUB dataset of birds (Wah et al., 2011) is shown in Table 3. Our model outperforms all previous methods, including UMR (Li et al., 2022) which leverages a pre-trained co-part segmentation network for supervision. We outperform CSM (Kulkarni et al., 2019) and A-CSM (Kulkarni et al., 2020) by over 20%, showing the efficacy of our CSE representation compared with the existing sur-

| Methods | KP | Mask | PCK |
|---|---|---|---|
| CMR (Kanazawa et al., 2018) | ✓ | ✓ | 47.3 |
| CSM (Kulkarni et al., 2019) | ✓ | ✓ | 45.8 |
| CSM | - | ✓ | 36.4 |
| A-CSM (Kulkarni et al., 2020) | - | ✓ | 42.6 |
| IMR (Tulsiani et al., 2020) | - | ✓ | 53.4 |
| UMR (Li et al., 2022) | - | ✓ | 58.2 |
| SMR (Hu et al., 2021) | - | ✓ | 62.2 |
| Ours | - | ✓ | **64.5** |

Table 3: **Keypoint transfer result on CUB-200-2011.**

| Method | IOU$_{0.25}$ | IOU$_{0.5}$ | 5° 2cm | 5° 5cm | 10° 2cm | 10° 5cm |
|---|---|---|---|---|---|---|
| w/o correspondence | 70.1 | 32.0 | 15.3 | 20.5 | 23.6 | 34.0 |
| w/o surface embedding | 87.3 | 57.1 | 24.0 | 30.1 | 31.8 | 42.1 |
| w/o cross-instance and cross-time cycle loss | 89.9 | 63.0 | 30.9 | 32.8 | 36.4 | 42.8 |
| w/o depth loss | 89.8 | 52.6 | 13.0 | 14.7 | 27.4 | 35.6 |
| w/o deformation | 87.5 | 55.3 | 21.2 | 26.8 | 32.8 | 44.0 |
| Full model | **92.3** | **68.2** | **32.7** | **35.3** | **38.3** | **45.3** |

Table 4: **Ablation study on model designs and loss functions.**

| Cycle level | | Laptop | | | | Bird |
|---|---|---|---|---|---|---|
| $\mathcal{L}_{\text{time}}$ | $\mathcal{L}_{\text{inst}}$ | IOU$_{0.5}$ | 5° 2cm | 5° 5cm | 10° 5cm | PCK |
| - | - | 94.6 | 9.2 | 10.2 | 38.8 | 33.9 |
| - | ✓ | 94.7 | 10.4 | 11.0 | 38.4 | 61.0 |
| ✓ | - | 95.9 | 11.6 | 13.5 | 39.7 | 63.4 |
| ✓ | ✓ | **96.0** | **12.7** | **16.0** | **42.9** | **64.5** |

Table 5: **Ablation study on the cross-instance and cross-time cycle-consistency loss.**

face mapping approach. We visualize the 2D-3D correspondence maps in Fig. 5. Despite a large shape and pose variance, our model can still estimate correct correspondence relationships across different images of birds.

### 4.3 ABLATION STUDY

**CSE and geometric correspondence.** We first evaluate the effectiveness of the proposed CSE representation and geometric correspondence. In *w/o correspondence* setting, we ablate both the mesh encoder and image feature extractor, and use no correspondence computation. At test time, instead of solving for pose from correspondences, we use the regressed rotation and translation from the image encoder. This results in a large performance drop, as shown in Table 4, which proves regressing pose leads to more difficulties in generalization than solving pose from correspondence.

In *w/o surface embedding* setting, we ablate the mesh encoder and have the image feature extractor directly output a surface mapping, which serves as the 2D-3D correspondence map. The performance under this setting is also significantly lower, showing the efficacy of our CSE representation.

**Cross-instance and cross-time cycle-consistency.** In our method, we propose the novel cross-instance and cross-time cycle-consistency loss to facilitate semantic-aware correspondence learning. Thus we also ablate this loss to verify its effect. In Table 4, we show the comparison training with and without the cross-instance and cross-time cycle-consistency loss on Wild6D. With the novel loss, performances at all metrics improve, with the mAP at IOU$_{0.5}$ improved by $5.2\%$. This proves that our novel cycle-consistency losses can effectively leverage image semantics and thus achieves an improvement over the single-image projection-based cycle loss.

In Table 5, we provide a more detailed ablation on the two cycle levels $\mathcal{L}_{\text{time}}$ and $\mathcal{L}_{\text{inst}}$. We take the laptop category as an example to show the effectiveness of the cross-instance and cross-time loss for it has large pose and appearance variances. From this table, we can observe that adding our proposed cycle-consistency loss constantly improves the pose estimation accuracy. Combining both cycle levels achieves the highest performance, and increases the mAP at 5 degrees, 5cm by $5.8\%$ in total. We also evaluate on the CUB dataset in Table 5. The PCK increases significantly by applying the cross-instance and cross-time cycle-consistency. The best-performing model increases over the model without cross-instance and cross-time cycle-consistency by $30.6\%$.

**Depth loss.** We also ablate the depth loss, shown in Table 4. Results show that using no depth data causes a performance drop. Empirically, we find that some categories are relatively depth-sensitive, e.g. bowl. Our depth loss can largely eliminate the misaligning of these shapes.

### 5 CONCLUSION

In this work, we consider the task of self-supervised category-level 6D object pose estimation in the wild. We propose a framework that learns the Categorical Surface Embedding representation and constructs dense geometric correspondences between images and 3D points on a categorical shape prior. To facilitate training of CSE and geometric correspondence, we propose novel cycle-consistency losses. In experiments, our self-supervised approach can perform on-par or even better than state-of-the-art methods on category-level 6D object pose estimation and keypoint transfer tasks.

**Reproducibility statement.** All experiments in this paper are reproducible. In Appendix A.1, we describe the detailed model architecture. In Appendix A.2 and Appendix A.3, we provide details for training and inference. In Appendix A.4, we provide all hyperparameters for training our model. The source code and training scripts are available at https://github.com/kywind/self-corr-pose.

**Acknowledgement.** Prof. Xiaolong Wang's group was supported, in part, by gifts from Qualcomm.

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

## A IMPLEMENTATION DETAILS

### A.1 MODEL ARCHITECTURE

**Image encoder.** We use ResNet-18 as our image encoder backbone. We apply average pooling on the output of the $4^{th}$ res-bloc, resulting in an image global feature with dimension 512. This

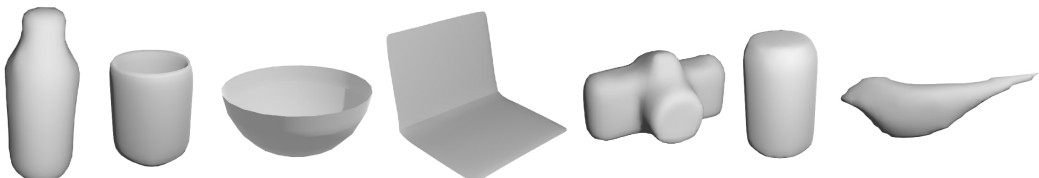

Figure 6: **Shape priors.** From left to right: bottle, mug, bowl, laptop, camera, can, bird.

| Properties | Bottle | Mug | Bowl | Laptop | Camera | Can | Bird |
|---|---|---|---|---|---|---|---|
| Dataset | W, R | W, R | W, R | W, R | W, R | R | CUB |
| # of vertices | 642 | 886 | 482 | 995 | 974 | 680 | 555 |
| Type of symmetry | rotation | flip | rotation | flip | - | rotation | flip |

Table 6: **Properties of each category.** W=Wild6D, R=REAL275.

global feature then serves as the input for the pose predictor, which uses a 4-layer MLP with hidden dimension 128 to regress the 6D rotation representation (Zhou et al., 2019), and a 1-layer translation predictor that predicts the 3D translation. The image global feature is also fed into a fully connected layer to get the shape code $\mathbf{u}_{shape}$ with dimension 64.

**Shape predictor.** We use a coordinate-based MLP with 5 fully connected layers for shape deformation prediction. The input for the network is the concatenation of the 3-dimensional vertex coordinate and the 64-dimensional global shape code, and the output is 3-dimensional $x, y, z$ offsets.

**Mesh encoder and image feature extractor.** We choose feature dimension 64 for our CSE representation. For the mesh encoder, we use a 3-layer PointNet (Qi et al., 2017) structure for extracting the surface embedding. For the image feature extractor, we use a PSPNet (Zhao et al., 2017) decoder which fuses different levels of the image encoder feature maps to get the final image features. The final feature map is of size $64 \times 64$ (4× downsampling).

## A.2 TRAINING DETAILS

**Data Sampling.** For each RGB image input, we crop the image to $256 \times 256$ and locate the object at the image center. In each training batch, we sample $b$ videos randomly from the dataset and sample $t$ frames from each video uniformly. For the CUB-200-2011 dataset, since the dataset contains no videos, we randomly sample $b$ bird species and $t$ images with each species instead. This sampling strategy ensures that there are multiple images from the same video/specie in a batch for cross-time cycle-consistency learning.

**Shape prior.** We select our shape prior by choosing a synthetic dataset from the ShapeNet (Chang et al., 2015) dataset, then simplify the mesh to contain an appropriate number of vertices. A visualization of all the shape priors we list is listed in Fig. 6. More properties of each category are listed in Table 6.

**Regularization.** We apply regularization losses in training, including deformation loss, Laplacian loss and symmetry loss. The deformation loss is the smooth L1 loss on the deformation $\Delta\mathbf{V}$. The Laplacian loss is by applying Laplacian smoothing on the deformed mesh. For symmetry loss, given a reconstructed mesh, we uniformly sample $m$ points, then apply $K$ rotate/flip transformations depending on symmetry types, creating a point cloud with $m \times K$ points. The symmetry loss is defined as the Chamfer distance between the point cloud and the original mesh.

**Details of instance cycle-consistency loss.** In the instance cycle-consistency loss (Eq. 4), we use $\pi^{-1}$ to denote the inverse projection operation. We implement this inverse projection by casting a camera ray starting from a pixel and finding its intersection with the mesh. The point becomes the inverse projection result of the pixel. The visibility mask of the instance cycle-consistency loss is applied over the vertices, with value 1 for the vertices that are visible after the projection, and value 0 for the vertices that are occluded.

**Details of cross-instance cycle-consistency loss.** When applying cross-instance cycle-consistency loss, we group batches of size $B$ into $B/2$ image pairs. For $\mathcal{L}_{inst}$, we pair images from different videos; for $\mathcal{L}_{time}$, we pair images from different frames of the same video. To stabilize training, we

| Hyperparameters | Wild6D | REAL275 | CUB |
|---|---|---|---|
| # of iterations | 20,000 | 10,000 | 5,000 |
| $(\beta_{\text{texture}}, \beta_{\text{mask}}, \beta_{\text{depth}})$ (Eq. 3) | $(0.05, 0.15, 0.1)$ | $(0.05, 0.15, 0.1)$ | $(0.05, 0.15, 0)$ |
| $(\beta_{\text{2D-3D}}, \beta_{\text{2D-3D}}, \beta_{\text{inst}}, \beta_{\text{inst}})$ (Sec. 3.3) | $(0.02, 0.02, 0.05, 0.05)$ | $(0.02, 0.02, 0.05, 0.05)$ | $(0.01, 0.01, 0.1, 0.1)$ |
| $\tau$ (Eq. 1) | 0.1 | 0.1 | 0.1 |
| $k$ (Eq. 6) | 200 | 200 | 200 |
| $(\lambda_{\text{recon}}, \lambda_{\text{cycle}}, \lambda_{\text{reg}})$ (Sec. 3.3) | $(1, 1, 1)$ | $(1, 1, 1)$ | $(1, 1, 1)$ |

Table 7: **Training hyperparameters.**

| Method | 3D/2D Transfer | PCK |
|---|---|---|
| VGG (Simonyan & Zisserman, 2014) | 2D | 17.2 |
| DINO (Caron et al., 2021) | 2D | 60.2 |
| Ours-2D | 2D | **72.9** |
| Ours | 3D | 64.5 |

Table 8: **Keypoint transfer result.** 2D Transfer indicates transferring keypoints directly with image-image feature distance. 3D Transfer indicates transferring keypoints with a mapping into the 3D space.

also add an auxiliary cycle loss, which is defined as follows: rotating the source image by a random degree to get a target image, and constructing a similar cycle as $\mathcal{L}_{\text{inst}}$. Correspondence pairs directly come from rotation.

**Depth.** For the category-level 6D pose estimation task, our model requires RGB-D image to train in order to achieve the best performance. The depth is only used for constructing a depth loss. During inference, depth is also required; otherwise, the relative scale of the translation cannot be correctly estimated due to the scale ambiguity in perspective projection. For the keypoint transfer task, our model only requires RGB images during both training and inference, since the task does not require to estimate the absolute depth of the object.

## A.3   INFERENCE DETAILS

**Keypoint transfer.** Given two images, we apply our image and mesh encoder to obtain the 2D-3D geometric correspondence of both images. Then, for each keypoint in the source image, we first extract its corresponding 3D location from the mapping. Then, for each pixel on the target mesh, we also map it into 3D. Finally, we choose the point with the closest 3D distance to the 3D location of the keypoint. The corresponding target pixel becomes the predicted keypoint. If the distance between our prediction and the ground truth is less than $\alpha \times \max(h, w)$ ($h, w$ are the height and width of the bounding box of the object), then the prediction is considered correct. In all experiments, we report PCK at $\alpha = 0.1$.

**Symmetry in pose estimation.** For rotation-symmetric categories (e.g. bottle, bowl, can), a correct prediction can be an arbitrary rotation of the ground truth pose along the symmetric axis. Therefore, we uniformly sample $K$ poses around the symmetric pose as candidate ground truth poses, and choose the one with the minimum error with the predicted pose.

## A.4   HYPERPARAMETERS

For all our experiments, we use AdamW (Loshchilov & Hutter, 2017) as our optimizer with learning rate $lr = 1 \times 10^{-4}$, and apply cosine learning rate decay. The learning rate for vertex deformation is set to $1 \times 10^{-5}$. We use batch size 64 (with 16 videos, 4 frames each video). More hyperparameters are listed in Table 7.

## A.5   WILD6D SEGMENTATION MASKS

We use the segmentation masks for Wild6D provided by Fu & Wang (2022). The segmentation is generated by applying Mask R-CNN (He et al., 2017). Although this auto-generation pipeline can produce some errors, we find that in most images the segmentation masks are satisfactory. Figure 7 shows some image and mask instances in the dataset.

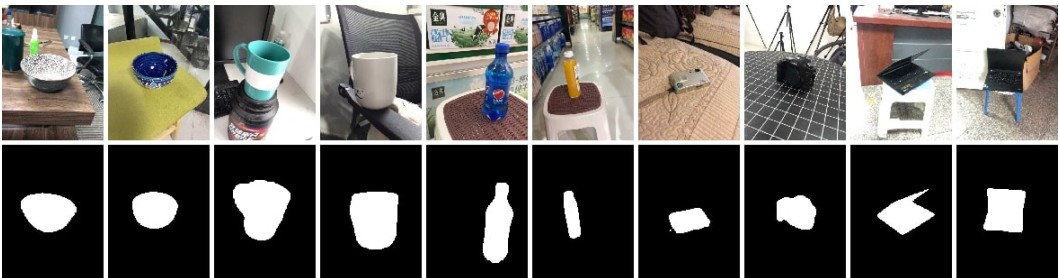

Figure 7: **Visualizing segmentation masks in Wild6D.**

# B    ADDITIONAL EXPERIMENT RESULTS

## B.1    COMPARISON WITH DINO ON KEYPOINT TRANSFER

In our work we use DINO (Caron et al., 2021), a self-supervised pre-trained image backbone, to find correspondence pairs between cross-instance images. To verify whether our model's performance gain solely relies on DINO, we conduct experiments to evaluate DINO on the task of keypoint transfer. The results are shown in Table 8. Compared with DINO, Ours-2D similarly uses direct 2D transferring, and results in higher performance than DINO (72.9% vs. 60.2%). Furthermore, our 3D-aware transferring method also outperforms DINO by 4.3%. These results show that our method is leveraging DINO correspondences to improve the feature representations, and we also learn 2D-3D correspondences, which DINO is not capable of.

# C    VISUALIZATION

We provide a more detailed visualization of our model on pose estimation, keypoint transfer, correspondence learning and mesh reconstruction. In Figure 8 and 9, we show the keypoint transfer result and the learned geometric correspondence on CUB-200-2011. In Figure 10, we provide some visualizations of the reconstructed meshes on Wild6D. In Figure 11, we show the correspondence and pose estimation results on REAL275. In Figure 12, we show comparisons of our estimated pose and the ground truth pose on the Wild6D dataset.

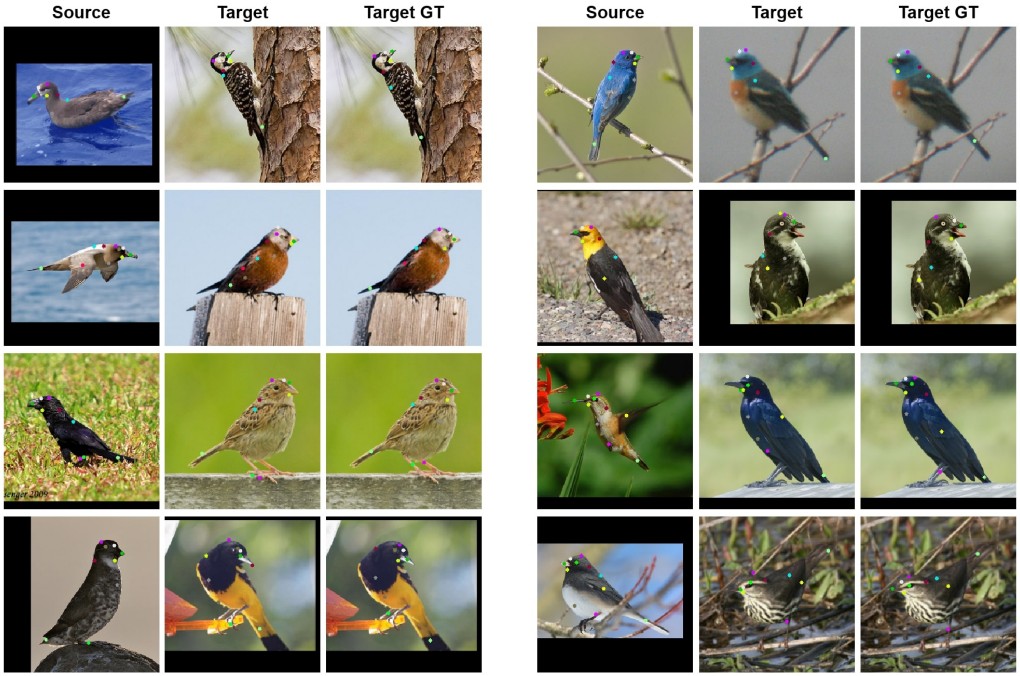

Figure 8: **Visualizing keypoint transfer on CUB-200-2011.** The three columns in a group represent the ground truth keypoints on the source image, the transferred keypoints on the target image, and the ground truth keypoints on the target image, respectively.

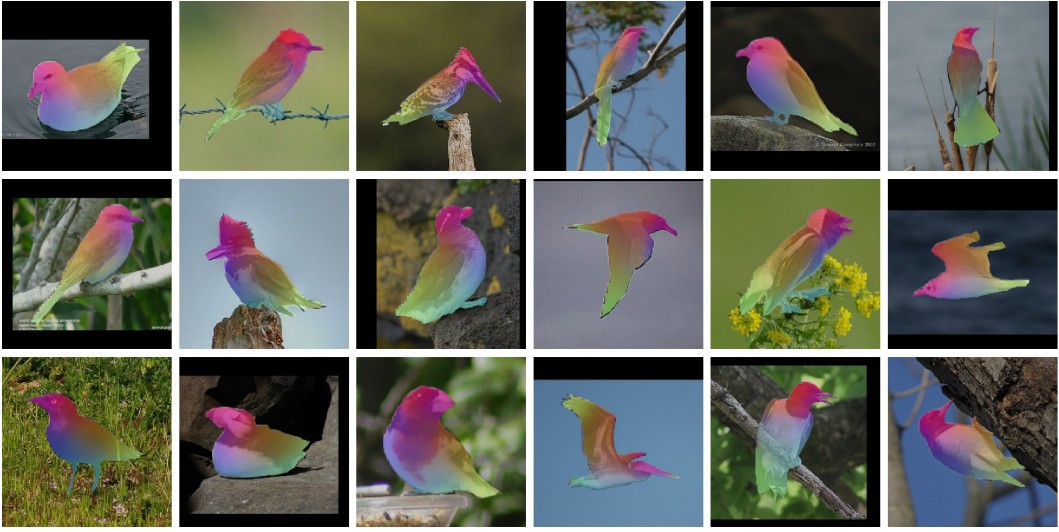

Figure 9: **Visualizing geometric correspondence on CUB-200-2011.** The color at each pixel depicts the corresponding 3D point location in the canonical space. Note that our model learns mostly continuous and consistent correspondences despite large pose and appearance changes.

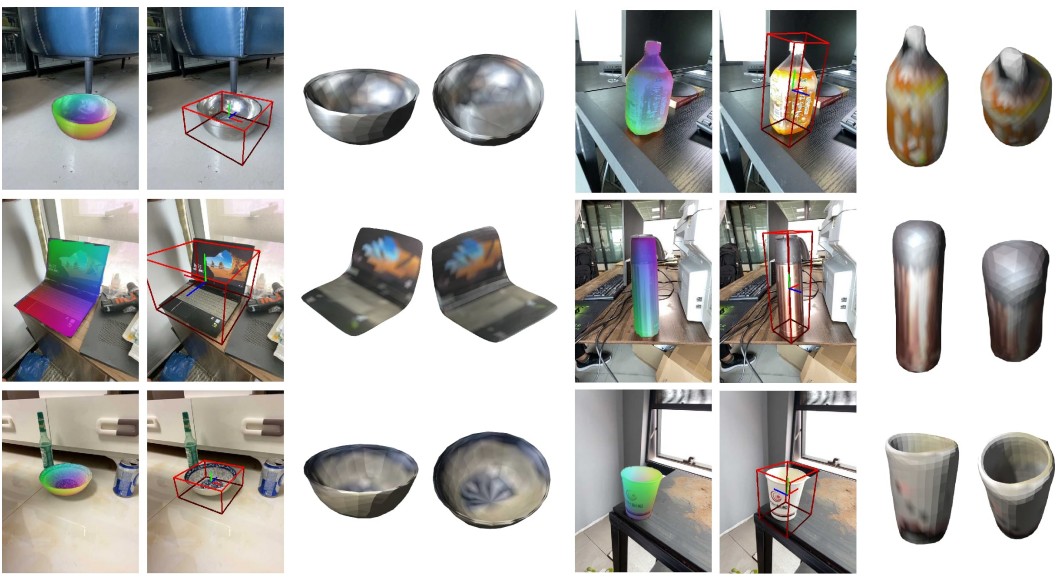

Figure 10: **Visualizing pose estimation and mesh reconstruction on Wild6D.** We show the correspondence mapping, the estimated pose and the reconstructed meshes under 2 different viewpoints chosen randomly.

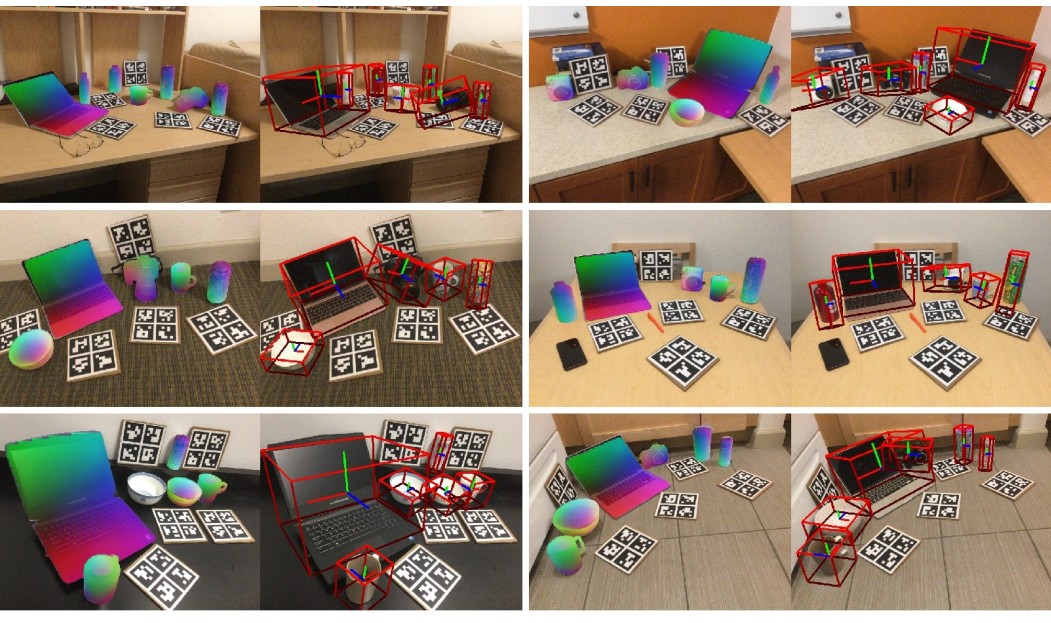

Figure 11: **Visualizing correspondence and pose estimation on REAL275.** We train the model only on Wild6D and test on REAL275, producing these results.

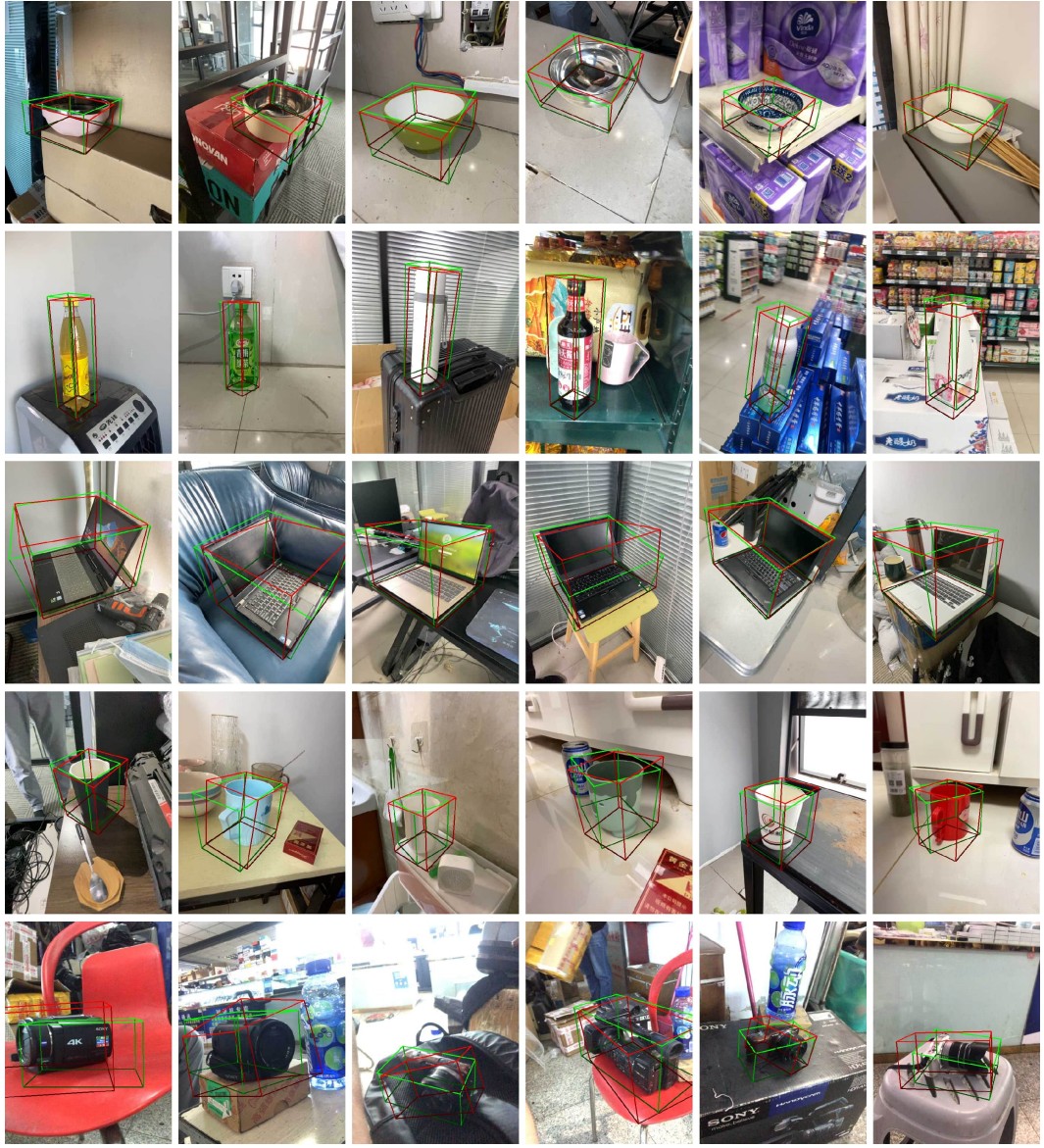

Figure 12: **Visualizing pose estimation on Wild6D.** The ground truth bounding boxes are colored in green, and the predicted bounding boxes are colored in red.

