# OpenReview forum: "Self-Supervised Geometric Correspondence for Category-Level 6D Object Pose Estimation in the Wild"
_ICLR.cc/2023/Conference — ICLR 2023 poster_

### Official Review · Reviewer_pKTW · 2022-10-21

**Confidence:** 4
**Clarity, Quality, Novelty And Reproducibility:** 1. In general, the paper is well writ…
**Correctness:** 3
**Technical Novelty And Significance:** 3
**Empirical Novelty And Significance:** 3
**Recommendation:** 6

**Strength And Weaknesses:**

Strength:
1. The proposed Categorical Surface Embedding module and the corresponding cycle-consistency loss function is interesting and novel.
2. The experiment result on pose estimation in the wild and dense key-point transfer on CUB dataset is impressive and the ablation study clearly shows the effectiveness of suface embedding and cycle loss.

Weakness:
1. There are some hyperparameters which weight different items in the overall loss function, such as \beta_texture, \beta_mask, \beta_2D-3D, etc. From Table 7 in Sec. A.3, it seems that such hyperparameters varies across Wild6D and CUB in training, does the algorithm's performance depends significantly on these hyper-parameters?
2. In Table 2, the proposed algorithm get good performance on the 'Self-supervised' setting, but significantly lower performance than algorithms in 'Synth supervised' setting, since the labeled synthetic images can be obtained with limited effort, what is the benefit of self-supervised training on Wild6D without pose label? and is it possible to test the setting of W*+R* data settting and compare to the method of UDA-COPE (Lee et al., 2022) with C+R* data setting?
3. In Figure 2, it is not clear whether and how the deformation \delta_V is regularized, there should be some sentences in sec.3.1 referring to Appendix A.2 and the regularization loss should be reflected in Figure 2.

**Summary Of The Paper:**

This paper proposed a self-supervised framework for category-level 3D object pose estimation. A novel category surface embedding module is proposed which can help establishing dense correspondences within a single instance, between two different instances of the same category, and instances in different time frames. Experiment shows that the proposed algorithm achieves impressive performance on 3D pose estimation and keypoint transfer.

**Summary Of The Review:**

In summary, I recommend that the paper could be accepted if all the questions in the weakness part are addressed

---

> ### Author Response · Authors · 2022-11-18
> **Author Response to Reviewer pKTW**
>
> Thank you for your detailed comments. We address each of the concerns as follows. Please also check the revised paper, with revisions colored in red.
>
> ---
>
> **Q**: “There are some hyperparameters which weight different items in the overall loss function, such as \beta_texture, \beta_mask, \beta_2D-3D, etc. From Table 7 in Sec. A.3, it seems that such hyperparameters varies across Wild6D and CUB in training, does the algorithm's performance depends significantly on these hyper-parameters?”
>
> **A**: The hyper-parameter choices can be dependent on the tasks. Within the same type of task, we use the same hyper-parameter across Wild6D and REAL275 for pose estimation besides training iterations. We use a different set of hyper-parameters for the cycle consistency losses on CUB since keypoint transfer requires more cross-instance consistency. To be specific, we increased the cross-instance cycle consistency weights and decreased the instance cycle consistency for balance. The depth weight is set to 0 since we do not use depth on CUB.
>
> ---
>
> **Q**: “In Table 2, the proposed algorithm get good performance on the 'Self-supervised' setting, but significantly lower performance than algorithms in 'Synth supervised' setting, since the labeled synthetic images can be obtained with limited effort, what is the benefit of self-supervised training on Wild6D without pose label? and is it possible to test the setting of W*+R* data setting and compare to the method of UDA-COPE (Lee et al., 2022) with C+R* data setting?”
>
> **A**: We believe the advantage of previous work using synthetic images comes from the designed synthetic images being well aligned with the real dataset REAL275. Such a design on a simulator might not necessarily be easy. While this is achievable when the real dataset is simple, the design of simulation to match the real world is much harder if we need to handle in-the-wild scenarios. It becomes more challenging to utilize the synthetic data if there is a large domain gap between the synthetic data and in-the-wild data.
>
> We have tried training with W*+R*; the performance is improved, but still lower than C+R* approaches. This is because that R* is very small (7 videos) compared with W*, and there is a large domain gap between them. Also, both R* and W* provides no labels, thus having more disadvantage compared to C+R*. In general, we find overfitting to the domain of R* with aligned synthetic data can reach a good result, but such an approach is not generalizable to in-the-wild settings and settings where the domain gap is not neglectable.
>
> (He et al., 2022) is a very recent work (ECCV 22) using no 3D annotations (not even from synthetic images). Under the same setting, our approach outperforms this work (Table 2).
>
> ---
>
> **Q**: “In Figure 2, it is not clear whether and how the deformation \delta_V is regularized, there should be some sentences in sec.3.1 referring to Appendix A.2 and the regularization loss should be reflected in Figure 2.”
>
> **A**: Thank you for pointing this out. We added descriptions of the regularization loss in Sec. 3.4, referring to Appendix A.2, and added the regularization loss in Fig. 2.

---

### Official Review · Reviewer_SV21 · 2022-10-23

**Confidence:** 4
**Correctness:** 3
**Technical Novelty And Significance:** 2
**Empirical Novelty And Significance:** 2
**Recommendation:** 6

**Clarity, Quality, Novelty And Reproducibility:**

The paper is overall clearly written but some important details need to be more crisp, such as requirement of depth data in input and if the mean shape is learned from scratch or taken from synthetic data.

Overall the paper has some degree of the novelty due to its unsupervised learning for 6D object pose. But majority of the component such as differentiable rendering, cycle-consistency loss are not new.

**Strength And Weaknesses:**

Strength
+ Overall the paper is clearly written and easy to follow.
+ The problem of 6D object pose estimation in the wild is of importance in practice, such as in robotic applications.
+ The major strength over prior arts lies in its unsupervised nature, which does not require any human annotation of object pose.
+ The experiments demonstrate state-of-the-art performance.

Weakness:
- Some of the technical deteils are not clear.  For example, it is not clear whether the mean shape is optimized or fixed - Sec 3.1 states that the mean shape is learnable but according to Sec. A.2 the mean shape is selected from synthetic data in ShapeNet.
- If the method learns both the category-level mean shape and per-instance shape deformation, the decompotion of mean shape and deformation is inherently ambiguious. Any change in the mean shape can be compansated by its inverse change in the deformation. It seems there is not any regularization to resolve such ambiguity.
- The paper has been vague about whether the depth is required in the input. My understanding is yes, because the method estimates pose by Umeyama algorithm. It is then a limitation if RGBD is required as input rather than RGB image only. The paper also acknowledges that removing depth loss leads to a drop in performance. Without depth data, how is the absolute scale of translation determined, given the ambiguity between objece size and distance caused by perspective projection?
- There is no ablation study on the impact of deformation compared to just using a canonical shape.
-  Many of the objects shown in the paper are symetric, hence it is again ambiguious to define the pose. Does this cause instabability in training and how is it handled?
 - The backprojection from 2D to 3D in the right equation of Eq.(4) needs depth as well. It is better to point out this for clarity.


**Summary Of The Paper:**

This paper proposes to learn category-level 6D object pose in an unsupervised manner. The key contribution lies in a framework to learn 2D-3D correspondences between image and a category-level canonical shape as a mesh. The correspondences are learned by regressing surface embedding on both image pixels and mesh vertices, which are then matched to obtain correspondences. The unsupervised losses mainly consist of differentiable rendering loss and various cycle-consistency losses.

**Summary Of The Review:**

The paper is solving a practically important problem, i.e. unsupervised learning of 6D object pose. The overall framework makes senses and works reasoablely in the experiments. That said, some important details needs further clarification as discussed above.

---

> ### Author Response · Authors · 2022-11-18
> **Author Response to Reviewer SV21 (2/2)**
>
> **Q**: “Many of the objects shown in the paper are symmetric, hence it is again ambiguous to define the pose. Does this cause instability in training and how is it handled?”
>
> **A**: We assume that the reviewer’s concern is whether the ambiguity of the pose will cause the estimated transformation to be unstable during training, which further causes unstable instance cycle consistency loss and harms correspondence learning.
>
> While the instability in training could happen theoretically, empirically we observe that the instance cycle-consistency loss makes the estimated transformation and the correspondence align with each other starting from a very early stage. The reason is: since both the transformation and the per-pixel features are predicted with a shared image encoder, the model can easily learn a correlation between them. Thus, they can align quickly during training with the effect of the instance cycle-consistency loss. As a result, the instability problem is prevented.
>
> ---
>
> **Q**: “The backprojection from 2D to 3D in the right equation of Eq.(4) needs depth as well. It is better to point out this for clarity.”
>
> **A**: In our implementation, Eq.(4) does not need depth. The back projection finds the intersection between a camera ray and the mesh. We made an explanation of this in the revised article, in appendix A.2.

---

> ### Author Response · Authors · 2022-11-18
> **Author Response to Reviewer SV21 (1/2)**
>
> Thank you for your detailed comments. We address each of the concerns as follows. Please also check the revised paper, with revisions colored in red.
>
> ---
>
> **Q**: Some of the technical details are not clear. For example, it is not clear whether the mean shape is optimized or fixed - Sec 3.1 states that the mean shape is learnable but according to Sec. A.2 the mean shape is selected from synthetic data in ShapeNet.
>
> **A**: The mean shape is initialized with a selected shape, then optimized during training. This is to stabilize training in the early phase and avoid the possible pose ambiguity brought by a sphere initialization.
>
> ---
>
> **Q**: “If the method learns both the category-level mean shape and per-instance shape deformation, the decomposition of mean shape and deformation is inherently ambiguous. Any change in the mean shape can be compensated by its inverse change in the deformation. It seems there is not any regularization to resolve such ambiguity.”
>
> **A**: We have applied a regularization loss to constrain shape deformation, avoiding such ambiguity. This is introduced in Appendix A.2 with the **Regularization** paragraph. Specifically, we applied an L1 loss to penalize the instance deformation, which encourages the learning of the mean shape. A similar technique is adopted in previous unsupervised category-level mesh reconstruction approaches, e.g. CMR (Kanazawa et al., 2018), and ViSER (Yang et al., 2021).
>
> ---
>
> **Q**: “The paper has been vague about whether the depth is required in the input. My understanding is yes, because the method estimates pose by Umeyama algorithm. It is then a limitation if RGBD is required as input rather than RGB image only. The paper also acknowledges that removing depth loss leads to a drop in performance. Without depth data, how is the absolute scale of translation determined, given the ambiguity between object size and distance caused by perspective projection?”
>
> **A**: We clarify that the depth information is indeed necessary during inference for 6D pose estimation. We agree with the reviewer that without using depth in inference, it will introduce ambiguity.
>
> During training time, our model does not take depth as input, but rather uses depth to provide supervision. This introduces some flexibility. For example, in our experiments with the CUB dataset, since it comes without depth, we train our model without using depth supervision and continue showing substantial improvement on the keypoint transfer task.
>
> We also conduct one experiment suggested by reviewer ohWa, where we train our model on the bottle category using estimated depth with pre-trained MiDaS (Ranftl et al., 2021) instead of using the ground-truth depth. We report the results in the following table. We find the results are close between using the estimated depth and GT depth for training. This result shows the potential of training our approach on RGB-only data. We would also like to emphasize that during testing with all approaches, we still adopt the GT depth provided in the test set for computing the pose using the Umeyama algorithm.
>
> | Method | IoU25 | IoU50 | 5deg 2cm | 5deg 5cm | 10deg 2cm | 10deg 5cm |
> | --- | --- | --- | --- | --- | --- | --- |
> | no depth |  89  |  68.7 |  48.3 |  53.1 |  68.9 | 77.9 |
> | estimated depth |  **89.8** | **79.9** | **68.7** | 73.9 | **76.6** |  84.2 |
> | GT depth | 89.4 |  78.9 |  68.2 |  **74** | 76.1 | **84.8** |
>
> ---
>
> **Q**: “There is no ablation study on the impact of deformation compared to just using a canonical shape.”
>
> **A**: *We have added the ablation of deformation in the revised article, in Table 4.* The following table shows our results. “No deform” means using only the mean shape without deformation. From the table, we see that ablating deformation lowers performance.
>
> | Method |  IoU25 | IoU50 | 5deg 2cm | 5deg 5cm | 10deg 2cm | 10deg 5cm |
> | --- | --- | --- | --- | --- | --- | --- |
> | no deform | 87.5 | 55.3 | 21.2 | 26.8 | 32.8 | 44.0 |
> | original | **92.3** | **68.2** | **32.7** | **35.3** | **38.3** | **45.3** |
>
> ---

---

### Official Review · Reviewer_ohWa · 2022-10-23

**Confidence:** 3
**Correctness:** 3
**Technical Novelty And Significance:** 3
**Empirical Novelty And Significance:** 2
**Recommendation:** 5

**Clarity, Quality, Novelty And Reproducibility:**

The paper is clearly written. There are some questions regarding the terminology used. In particular, (Umeyama 1991) algorithm is for the 3D-3D alignment problem, and it is incorrect to say that it is applied for the 2D-3D alignment problem.

Also, the authors call the competitive papers 'supervised or semi-supervised', while the authors of those papers call their approaches 'self-supervised'. It relates, for example, to the work (He et al., 2022), (Peng et al., 2022), RePoNet. Would like to understand clearly why does this happen. In my opinion, those works also fall into 'self-supervised' category.




**Strength And Weaknesses:**

I see at least three strong points in the submitted work. First of all, the idea to formulate 2D-3D pose regression through cross-attention between the image pixel features and the mesh vertex vertices seems new, original and interesting. It is one step forward compared to the direct regression of 3D coordinates from 2D images, which became a traditional tool in dense correspondence and pose estimation models wince the Vitruvian Manifold work of 2012. As the ablation study shows, without this element, the model starts to perform significantly worse.

The second strong point is that the evaluation is done on a big real dataset of objects (1722 videos of 5 object categories). The baselines are chosen as rather recent methods, and the metrics are standard and widely used.

The third strong point is the evaluation on a keypoint correspondence task, which shows the power of the learned model beyond the primary task.

However, while the current dataset is diverse in terms of objects, it is very constrained in terms of object categories (only 5 of them, and all are rather simple). It is unclear whether a method that performs well on these categories will generalize to arbitrarily complex categories, e.g. a vehicle, etc., and there is a very popular and recent dataset (CO3D) that can be used as well.

Next, it is not well described in the paper, in my opinion, that the method needs depth data to train. It would be very interesting to see what metrics can we have if we don't use depth for training. It is often the case that the dataset of RGB images is much easier to collect. As an intermediate baseline, maybe one may use some self-supervised depth estimation models here as well. I see no problem in relying on some machine learning models for annotation if they are self-supervised as well.

While I see that the reconstruction loss is important, probably I'd suggest explaining clearly why is it needed if we have the instance consistency loss. The problem is, the reconstruction loss enforces the colour consistency constraint between the projected vertices and the pixels. The instance consistency loss basically enforces consistency between the projected vertices and the pixels as well. It is stronger, because colours may be the same or similar, but coordinates may be very different. It is also not very clear why removing the depth loss makes things worse, because in my opinion (correct me if I'm wrong) but the depth supervision is already inside the instance consistency loss.

The experiment with keypoint transfer is interesting. But I would like to see the result of keypoint transfer by DINO features as well, to be able to compare and understand if there are any gains compared to this model.

Moreover, there is one question. Why are tthe DINO features used in the cycle consistency, but not the proposed 2D features?

In the Fig. 2, the cycle losses should be shown as well.





**Summary Of The Paper:**

The paper describes to learn a model for category-level object pose estimation. The proposed approach relies on a dataset of RGB-D images of a certain category of objects, e.g. mugs or bottles, with a template mesh of an object belonging to the category. The method does not require any pose annotation to train, however it leverages recent DINO features during training. The model can still be called self-supervised because DINO learning is self-supervised itself. The approach is evaluated on the Wild6D dataset and is shown to be on par with the current state-of-the-art.

**Summary Of The Review:**

Summarising, the paper contains good and valuable ideas. The evaluation shows, that the method is the best on Wild6D in terms of IoU .25 metric as well as pose metrics. However, the situation is mixed on the REAL275 dataset, and here there are question regarding the adequate baselines, because the paper refuses to accept that the competitive methods also fall into the 'self-supervised' category, while the competitive methods are also called 'self-supervised'. This is a difficult situation. Also, the paper is evaluated on just one in-the-wild dataset with 5 simple object categories. I would suggest evaluation on a ore diverse set of object categories.

---

> ### Author Response · Authors · 2022-11-18
> **Author Response to Reviewer ohWa (2/2)**
>
> **Q**: “The experiment with keypoint transfer is interesting. But I would like to see the result of keypoint transfer by DINO features as well, to be able to compare and understand if there are any gains compared to this model.”
>
> **A**: This result is already available in Table 8 in the appendix. We also list the results in the following table, showing our method has significant gain over the DINO feature.
>
> | Method |Transfer type | PCK @ 0.1 |
> | --- | --- | --- |
> | DINO |  2D-2D | 60.2 |
> | Ours | 2D-2D | **72.9** |
> | Ours | 2D-3D-2D | **64.5** |
>
> Transfer type 2D-2D means the 2D feature similarity is used for transferring, and 2D-3D-2D means the standard approach of using the 2D-3D correspondence to transfer.
>
> ---
>
> **Q**: “Moreover, there is one question. Why are the DINO features used in the cycle consistency, but not the proposed 2D features?”
>
> **A**: This will lead to shortcuts. We use feature similarity to find correspondence, and if we construct the cycle using the same set of features, then the cross-time and cross-instance cycle consistency will degenerate into the instance cycle consistency.
>
> ---
>
> **Q**: “In Fig. 2, the cycle losses should be shown as well.”
>
> **A**: The cycle consistency has been partially shown in Fig.2, denoted as $\mathcal{L}_{\text{cycle}}$. It is drawn between the estimated transformation and the correspondence to show that the loss depends on these terms. The details of the cycle loss is shown in Fig.3.
>
> ---
>
> **Q**: “There are some questions regarding the terminology used. In particular, (Umeyama 1991) algorithm is for the 3D-3D alignment problem, and it is incorrect to say that it is applied for the 2D-3D alignment problem.”
>
> **A**: Thank you for pointing out this. We apply the Umeyama algorithm in a two-step manner: lift the 2D RGB-D image to 3D; then apply the 3D-3D alignment for the pose. *This is clarified in the revised paper, section 3.4.*
>
> ---
>
> **Q**: “Also, the authors call the competitive papers 'supervised or semi-supervised', while the authors of those papers call their approaches 'self-supervised'. It relates, for example, to the work (He et al., 2022), (Peng et al., 2022), RePoNet. Would like to understand clearly why does this happen. In my opinion, those works also fall into 'self-supervised' category.
> and here there are question regarding the adequate baselines, because the paper refuses to accept that the competitive methods also fall into the 'self-supervised' category, while the competitive methods are also called 'self-supervised'. This is a difficult situation.”
>
> **A**: It is true that the use of the ‘self-supervised’ term is ambiguous across different papers. We respect the naming of previous approaches and modify our paper accordingly (see Table 2 in the revised paper). However, above all, we would like to clarify that our method tackles the problem in the **most challenging setting**: without using synthetic data, and without using real-world 3D annotations. This is the first work that achieves in-the-wild pose estimation under this setting, and the results are **comparable or even better** than the approaches using more annotations. We believe this strong result has great value.
>
> Some previous approaches using the synthetic data and annotations are called ‘self-supervised’, given no human labeling is required. We believe there are still two main limitations if the approach relies heavily on synthetic data: (i) The sim2real gap, extra efforts on domain adaptation will be required to well utilize the synthetic data; (ii) Most approaches are designing the synthetic data and environment given a presented real dataset. While this is achievable when the real dataset is simple, the design of simulation to match the real world is much harder if we need to handle in-the-wild scenarios. Given these limitations, we believe developing a method that does not need to rely on simulation provides a promising way to tackle in-the-wild problems.

---

> > ### Comment · Reviewer_ohWa · 2022-11-30
> > **Reviewer response**
> >
> > Thank you for the detailed response.
> >
> > First of all, I see how depth is used, and I agree that CO3D does not have proper metric scale. However, it is still possible to have some results, up to scale. Maybe here we should all agree that it goes beyond the scope of the current submission.
> >
> > Now I see that instance consistency and reconstruction losses can work together.
> >
> > Finally, the experiments show that on the Wild6D data, the method is undoubtedly superior compared to the baselines. This is a very good empirical contribution.
> >
> > All in all, I lean towards acceptance of the paper after the modifications and clarifications made by the authors.

---

> > > ### Author Response · Authors · 2022-12-07
> > > **Thank you!**
> > >
> > > Dear Reviewer,
> > >
> > > Thank you so much for your positive feedback. We appreciate the time and effort you have put into reviewing our modifications and responses. And we are glad that the questions have been resolved!

---

> > > ### Author Response · Authors · 2022-12-09
> > > **Rating**
> > >
> > > Dear reviewer,
> > >
> > > Would you consider increasing the rating so it will be easier for the AC to do the final recommendation?
> > >
> > > Thank you so much again!
> > >
> > > Authors.

---

> ### Author Response · Authors · 2022-11-18
> **Author Response to Reviewer ohWa (1/2)**
>
> Thank you for your detailed comments. We address each of the concerns as follows. Please also check the revised paper, with revisions colored in red.
>
> **Q**: “While the current dataset is diverse in terms of objects, it is very constrained in terms of object categories (only 5 of them, and all are rather simple). It is unclear whether a method that performs well on these categories will generalize to arbitrarily complex categories, e.g. a vehicle, etc., and there is a very popular and recent dataset (CO3D) that can be used as well.”
>
> **A**: We are already adopting the **largest** RGB-D object datasets. For the task of category-level 6D object pose estimation, commonly used datasets are Wild6D (5 categories) and NOCS (6 categories) datasets, containing the most representative household items, with large shape variance within each category. Specifically, the recently proposed Wild6D dataset (NeurIPS’22) is **one of the largest in-the-wild RGBD** object-centric datasets. To keep consistency with previous works, we directly adopt these datasets.
>
> While there are also datasets such as CO3D with more categories, the dataset **does not come with GT depth images**, which makes it not suitable for the task of 6D pose estimation. Note that without GT depth information in testing, it can easily create scale ambiguity during pose estimation: Confusion between if the object is smaller or placed further away.
>
> ---
>
> **Q**: “it is not well described in the paper, in my opinion, that the method needs depth data to train. It would be very interesting to see what metrics can we have if we don't use depth for training. It is often the case that the dataset of RGB images is much easier to collect. As an intermediate baseline, maybe one may use some self-supervised depth estimation models here as well. I see no problem in relying on some machine learning models for annotation if they are self-supervised as well.”
>
> **A**:  We first clarify that, during training time, our model only takes an RGB image as the input, and uses depth to provide one of the supervision (loss). We indeed ablate the model without using depth to provide the supervision in Table 4 in the paper. For the CUB dataset, since it comes without depth, we also train our model without using depth supervision and continue to show substantial improvement on the keypoint transfer task.
>
> Per the reviewer’s suggestion, we have also trained our model on the bottle category using estimated depth with pre-trained MiDaS (Ranftl et al., 2021) instead of using the ground-truth depth. We report the results in the following table. We find the results are close between using the estimated depth and GT depth for training. We would also like to emphasize that during testing with all approaches, we still adopt the GT depth provided in the test set for computing the pose using the Umeyama algorithm.
>
> | Method | IoU25 | IoU50 | 5deg 2cm | 5deg 5cm | 10deg 2cm | 10deg 5cm |
> | --- | --- | --- | --- | --- | --- | --- |
> | no depth | 89 | 68.7 |  48.3 |  53.1 |  68.9 | 77.9 |
> | estimated depth |  **89.8** | **79.9** | **68.7** | 73.9 | **76.6** |  84.2 |
> | GT depth | 89.4 |  78.9 |  68.2 | **74** |  76.1 | **84.8** |
>
> *We have clarified the description related to how depth is utilized in the revised paper*.
>
> ---
>
> **Q**: “While I see that the reconstruction loss is important, probably I'd suggest explaining clearly why is it needed if we have the instance consistency loss. The problem is, the reconstruction loss enforces the colour consistency constraint between the projected vertices and the pixels. The instance consistency loss basically enforces consistency between the projected vertices and the pixels as well. It is stronger, because colours may be the same or similar, but coordinates may be very different. It is also not very clear why removing the depth loss makes things worse, because in my opinion (correct me if I'm wrong) but the depth supervision is already inside the instance consistency loss.”
>
> **A**:  These losses are complementary to each other. The mask reconstruction loss does not directly utilize the 2D-3D correspondence, but it optimizes both object shape and poses with back-propagation. The instance cycle-consistency loss directly optimizes the 2D-3D correspondence, based on reasonable shape and pose estimation. The texture loss optimizes all the components together. The depth reconstruction loss is also useful: the depth supervision is **not** in the instance consistency loss, as we do not use depth in constructing the inverse projection; instead, we find the intersection between a camera ray and the mesh as the inverse projection point, thus it does not contain depth supervision. *We have updated the description of this point in the paper, in Sec. 3.3 and Appendix A.2*.
>
> ---

---

### Official Review · Reviewer_hRgU · 2022-10-25

**Confidence:** 3
**Correctness:** 4
**Technical Novelty And Significance:** 3
**Empirical Novelty And Significance:** 2
**Recommendation:** 8

**Clarity, Quality, Novelty And Reproducibility:**

The paper is clearly written, probably reproducible, and to my knowledge is novel.

**Strength And Weaknesses:**

The self-supervised nature of the method is attractive and makes it more applicable due to lower costs of dataset acquisition. The results improve the SOTA. Interestingly, the paper achieves better results than some supervised methods. Ablation study validates the architecture and loss choices. The methods is validated on two domains (images and videos) and two tasks (6d pose estimation itself and key points transfer)

**Summary Of The Paper:**

The paper proposes a self-supervised method for Category-level 6D Object Pose Estimation. Specifically, the paper proposes an architecture and a number of geometry-based consistency losses allowing simultaneous estimation of the shape of specific instance and its pose. The experimental sections, demonstrates improvements over SOTA.

**Summary Of The Review:**

I give positive rating because in my opinion the paper is well written and novel, and will be interesting to the community.

---

> ### Author Response · Authors · 2022-11-18
> **Author Response to Reviewer hRgU**
>
> Thank you for your encouraging comments and for pointing out several strong points of our paper! We hope that our paper will be interesting and contribute to the community. We also addressed the concerns of other reviewers and revised our article to make several additional experiments and clarifications.

---

### Author Response · Authors · 2022-11-18
**General Response**

Dear reviewers, we appreciate all the detailed comments and helpful suggestions. We have highlighted the changes in red in the revised version of our paper. Specifically, we have made the following updates:

- We have added the ablation of deformation to answer Reviewer SV21’s question. Results show that ablating deformation will lower the performance.

- We have clarified the requirement of depth, and added a more detailed description of the pose fitting method, following Reviewer ohWa and Reviewer SV21’s comments.

- We have replaced the terms ‘synth-supervised’ and ‘self-supervised’ with ‘self-supervised (with synthetic data and annotations)’ and ‘self-supervised (without synthetic data)’, to keep consistency with their original papers. A more detailed clarification of the synthetic data and the self-supervised term is given below.

- We have added descriptions of the regularization loss in both Fig.2 and section 3.4, with a reference to the appendix following Reviewer pKTW’s suggestions.

- We have revised the description of the shape prior in section 3.1 following Reviewer SV21’s comments.

- We have revised the description of the relationship between the reconstruction loss and the cycle consistency loss in section 3.2 following Reviewer ohWa’s comments.

- We have added more details of the instance cycle consistency loss in Appendix A.2 following Review SV21’s comments.

---

**About using synthetic data and the ‘self-supervised’ term**

It is true that the use of the ‘self-supervised’ term is ambiguous across different papers. We respect the naming of previous approaches and modify our paper accordingly (see Table 2 in the revised paper). However, above all, we would like to clarify that our method tackles the problem in the **most challenging setting**: without using synthetic data, and without using real-world 3D annotations. This is the first work that achieves in-the-wild pose estimation under this setting, and the results are **comparable or even better** than the approaches using more annotations. We believe this strong result has great value.

Some previous approaches using the synthetic data and annotations are called ‘self-supervised’, given no human labeling is required. We believe there are still two main limitations if the approach relies heavily on synthetic data: (i) The sim2real gap, extra efforts on domain adaptation will be required to well utilize the synthetic data; (ii) Most approaches are designing the synthetic data and environment given a presented real dataset. While this is achievable when the real dataset is simple, the design of simulation to match the real world is much harder if we need to handle in-the-wild scenarios. Given these limitations, we believe developing a method that does not need to rely on simulation provides a promising way to tackle in-the-wild problems.

---

### Public Comment · ~Cristóvão_D._Sousa1 · 2023-10-05
**Question regarding correspondence matrices and correspondence mapping equations**

Hi! In equation (1) shouldn't $u$ be indexed by $j$ and $v$ be indexed by $i$?
That would make more sense for the computations of the 2D-3D matrix where the exponential cosine distance between a pixel to a given vertex is normalized (divided) by the sum of distances of that pixel to all vertices. And the same for the 3D-2D matrix.
Also, that way it would be consistent with equation (2) where $i$ ranges over all $N$ and $j$ ranges over all $h×w$.

---

### Decision · Program_Chairs · 2023-01-20

**Decision:**

Accept: poster

**Justification For Why Not Higher Score:**

The paper would be stronger if it had relaxes the RGBD assumption so that it can be compared to other datasets like objectron. This is not a strong criticism for rejecting the paper but relaxing this assumption would make it applicable to a lot more broader audience.

**Justification For Why Not Lower Score:**

Self supervised 6dof prediction is an important open problem in computer vision and robotics. This is the first paper to demonstrate good results on a real world 6dof dataset, with a reasonable baselines and formulation.

**Metareview: Summary, Strengths And Weaknesses:**

Current 6dof mesh pose predictors are either trained with supervised learning or 3d simulators. There is little work in self supervised learning in this particular domain. The authors present an architecture to take RGB-D images and a template mesh of the considered object categories, which is then used to learn a surface embeddings to map the 3D surface to corresponding pixel values. A geometric cycle consistency loss maps the 2D to 3D. The novelty is that this approach does not require human labeling and seems to be the first result on a large enough 6dof dataset (Wild6D).

This paper is tackling an open research problem and presents a sound solution. The writing is reasonable and clear enough. The biggest weakness is that it dosen't yet compare to all the datasets, especially things like CO3D or objectron. However, it is not fair to judge the paper because those datasets do not provide RGBD information which is needed for this approach. I do not think RGBD is a big limitation and it is interesting and realistic enough setting to consider and work on.

I also think that most of the issues brought up during the discussion phase were resolved by the authors.

**Note From Pc:**

if the above contains the word "oral" or "spotlight" please see: "oral" presentation means -> notable-top-5% and "spotlight" means -> notable-top-25%. As stated in our emails, we are disassociating presentation type from AC recommendations